# A Case Study of Successful Application of the Principles of ME/CFS Care to an Individual with Long COVID

**DOI:** 10.3390/healthcare11060865

**Published:** 2023-03-16

**Authors:** Lindsay S. Petracek, Camille A. Broussard, Renee L. Swope, Peter C. Rowe

**Affiliations:** The Division of Adolescent and Young Adult Medicine, Department of Pediatrics, Johns Hopkins University School of Medicine Baltimore, Baltimore, MD 21205, USA; lpetrac1@jhmi.edu (L.S.P.); camille.broussard@jhmi.edu (C.A.B.); rswope3@jhmi.edu (R.L.S.)

**Keywords:** post COVID-19 condition, long COVID, myalgic encephalomyelitis, chronic fatigue syndrome, orthostatic intolerance, adolescent

## Abstract

Persistent fatigue is one of the most common symptoms of post-COVID conditions, also termed long COVID. At the extreme end of the severity spectrum, some individuals with long COVID also meet the criteria for myalgic encephalomyelitis/chronic fatigue syndrome (ME/CFS), raising the possibility that symptom management approaches for ME/CFS may benefit some long COVID patients. We describe the long-term outcomes of a 19-year-old male who developed profound impairment consistent with ME/CFS after a SARS-CoV-2 infection early in the pandemic. We evaluated and treated him using our clinic’s approach to ME/CFS. This included a history and physical examination that ascertained joint hypermobility, pathological reflexes, physical therapy maneuvers to look for a range of motion restrictions in the limbs and spine, orthostatic testing, and screening laboratory studies. He was found to have profound postural tachycardia syndrome, several ranges of motion restrictions, and mast cell activation syndrome. He was treated according to our clinic’s guidelines for managing ME/CFS, which included manual physical therapy maneuvers and both non-pharmacologic measures and medications directed at postural tachycardia syndrome and mast cell activation. He experienced significant improvement in his symptoms over 30 months. His case emphasizes how the application of the principles of treating ME/CFS has the potential to provide a direction for treating long COVID.

## 1. Introduction

Soon after the onset of the coronavirus pandemic in 2020, it became evident that fatigue was a common symptom during acute SARS-CoV-2 infections and that it could accompany mild or even asymptomatic respiratory illness [1]. Persistent fatigue has emerged as one of the most prominent symptoms of post-COVID conditions [2], also termed long COVID [3,4]. At the more extreme end of the spectrum of impairment, some individuals with long COVID meet the diagnostic criteria for myalgic encephalomyelitis/chronic fatigue syndrome (ME/CFS) [5,6,7]. The risk factors for developing long COVID are emerging but are incomplete at this time, although female sex is a common risk factor for both ME/CFS and long COVID [8,9,10]. The optimal treatment for post-COVID conditions remains to be determined. As mentioned in the US Centers for Disease Control and Prevention (CDC) guidance [4], the overlap of symptoms between long COVID and ME/CFS, dysautonomia, and mast cell activation syndrome, among others, suggests that symptom management approaches helpful for these disorders may benefit some patients with post-COVID conditions.

The Institute of Medicine criteria for the diagnosis of ME/CFS require (1) a substantial impairment in previously tolerated activities (usually accompanied by profound fatigue), (2) post-exertional malaise, (3) unrefreshing sleep, and either (4a) cognitive impairment or (4b) orthostatic intolerance. The symptoms should be present for at least six months, for at least half the time, and with at least moderate intensity [11]. We describe the long-term follow-up of a patient who met the Institute of Medicine criteria for ME/CFS at six months following a confirmed acute SARS-CoV-2 infection. We reported his short-term history after infection in an earlier case series [6] and here describe in greater detail his response over 30 months, emphasizing how the application of the lessons learned in managing ME/CFS informed the treatment of his long COVID.

## 2. Evaluation

In our clinic, we evaluate patients with possible ME/CFS using the following elements: (1) history (supplemented by questionnaires), (2) a physical examination with a careful neurological examination, a Beighton score for joint hypermobility, and physical therapy maneuvers to look for a restricted range of motion of the limbs and spine, (3) orthostatic testing, and (4) screening laboratory studies [12,13,14,15,16,17,18].

Case Report

A 19-year-old male with a history of allergic inflammation developed a coronavirus infection in June 2020, before the availability of SARS-COV-2 vaccines. His diagnosis was confirmed by a SARS-CoV-2 nucleic acid amplification test. Given an infection occurring early in the pandemic, this was likely to have been the historical variant of the virus. He experienced mild respiratory symptoms and anosmia but was never hypoxic and did not require hospitalization. As the acute symptoms were lifting two weeks after their onset, he noticed an elevation in his heart rate while walking between rooms in his house. Over the next two months, he was unable to exercise without provoking post-exertional malaise. As an example, after a game of corn hole (bean bag toss), he noted an elevation in heart rate up to 170 beats per minute (bpm) for approximately 30 min, followed by three days of needing to rest in bed. Extensive cardiovascular testing revealed no abnormalities except for a mildly reduced peak oxygen uptake of 84% of predicted during cardiopulmonary exercise testing.

Two months after the onset of his infection, his main symptoms were constant fatigue, unrefreshing sleep, lightheadedness, post-exertional malaise after mild increases in activity, bi-frontal and bi-temporal headaches, occasional cough, chest pain, leg pain, and mild anxiety and depression. We use a unidimensional wellness score to evaluate overall function [19]. This score asks how the individual rates his or her health over the preceding month, with responses anchored at 0 for dying and 100 for the best an individual could feel. Before the illness, he described an overall wellness score of 100, and at the 2–3 month point, his wellness score was 45, consistent with a moderate to severe impairment in function.

### 2.1. Comments: Physical and Neurological Examination

Because 60% of adolescent and young adult ME/CFS patients in our studies meet the Beighton score criteria for joint hypermobility (defined as a score of 4 or higher), compared to only 24% of healthy controls, we obtain a Beighton score in all patients. The odds ratio for having joint hypermobility in an adolescent or young adult with ME/CFS is 3.5 (95% confidence interval 1.6–7.5) [20]. 

Other work has shown high rates of biomechanical dysfunction in those with ME/CFS [12,21]. After controlling for the degree of joint hypermobility, ME/CFS patients are more likely to have a restricted range of motion of the limbs and spine than are healthy controls on an 11-point range of motion (ROM) score in which higher scores reflect worse dysfunction [20]. The median ROM score for patients is 5, compared to 2 for healthy controls (*p* < 0.001). As a result, we include in our physical examination a number of simple measures of range of motion that are usually part of a physical therapist’s evaluation (seated slump testing, ankle dorsiflexion, passive straight leg raise, brachial plexus strain testing, prone knee bend, and prone press up). 

A subset of ME/CFS patients has neuroanatomic abnormalities including Chiari I malformation, cervical spinal stenosis, and craniocervical or atlantoaxial instability [12,17,20,22,23,24]. We, therefore, examine carefully for brisk reflexes and assess for the presence of a Hoffman sign [14,25] in all. 

Case Report

His Beighton score was 3/9. He had 2+ symmetrical deep-tendon reflexes but a bilaterally positive Hoffman sign. Examination of his range of motion revealed limitations in several areas. On seated slump testing, which involves neck and spinal flexion followed by unilateral leg extension, he lacked 30 degrees of full leg extension on the right and 60 degrees on the left. His passive straight leg raise reached end-range at 35 degrees bilaterally which was restricted for his degree of overall flexibility. Upper limb neurodynamic testing with a median nerve bias elicited stretch across the antecubital fossa at 130 degrees of elbow extension on the left and 140 degrees on the right (normal is >170 degrees of elbow extension). A cervical spine MRI showed no cervical stenosis or cerebellar tonsillar descent below the foramen magnum. 

### 2.2. Comments: Orthostatic Testing

Over 95% of adolescent and young adult ME/CFS patients exhibit evidence of orthostatic intolerance [11,26]. We screen for this in the clinic using an inexpensive, ten-minute passive standing test in all who are being evaluated for ME/CFS, even if they are not endorsing lightheadedness [27,28]. After five minutes supine, patients stand quietly for ten minutes, leaning their shoulders against the wall, with instruction to avoid fidgeting and shifting their weight. 

Case Report

The passive standing test showed a 70 bpm difference between his lowest supine and peak standing heart rate, along with an increased headache, consistent with postural tachycardia syndrome (POTS) (Figure 1). The heart rate increment required for the diagnosis of POTS at age 19 is 40 bpm between the lowest supine and the peak value upright [29].

### 2.3. Comment: Laboratory Studies

We recommend the following screening laboratory assessments in those with chronic fatigue: complete blood count (CBC) with differential white blood cell count (WBC), erythrocyte sedimentation rate (ESR) or C-reactive protein (CRP), serum chemistry panel, free T4, thyroid stimulating hormone (TSH), urinalysis, iron studies (ferritin or iron/total iron binding capacity), vitamin B12, and celiac screening. Other studies are obtained as clinically indicated (for example, histamine, chromogranin A [in those not on proton pump inhibitors], tryptase, and other mast cell mediators in the presence of allergic and mast cell symptoms [13]). The patient’s laboratory studies are presented in Table 1.

## 3. Treatment

Our management approach begins with an explanation of the hypothesized pathophysiology of symptoms and a description of the usual prognosis of ME/CFS, emphasizing that many patients experience improvement with multi-modal therapy directed at their symptoms, although such improvement is not universal. We then develop working hypotheses about the primary influences on fatigue and overall function, beginning non-pharmacologic therapies directed at those influences and moving relatively rapidly to pharmacologic therapy as indicated. We reassess at regular intervals of one to two months, at each interval examining which influences on symptoms need the greatest attention.

In this young man, the hypothesized primary influences on his major symptoms were the profound postural tachycardia syndrome, the allergic inflammation, the elevated histamine and chromogranin A that were suggestive of mast cell activation syndrome [30], the increase in sensory sensitivity, and his reduced range of motion on physical therapy screening tests.

### 3.1. Comment: Treatment of the Orthostatic Intolerance 

This usually involves recommending the non-pharmacologic measures of increasing fluid intake, aiming for at least 2 L of oral fluids per day, increasing the intake of high sodium foods, liberal use of the salt shaker, and, if needed, using salt tablets and electrolyte packets. We encourage the use of compression garments (stockings, abdominal compression, and body shaper garments), usually at 20–30 mm Hg for stocking compression. Some patients tolerate 30–40 mm Hg leg compression, but those with joint hypermobility or extreme fatigue may find these too difficult to put on and take off. We suggest postural counter-maneuvers that engage the leg muscles to improve venous return. Examples of the latter include standing with the legs crossed, shifting weight when standing, sitting with the knees elevated above the hips (for example, resting the feet on a knapsack or camping stool), or leaning forward when sitting [31]. 

We then move, usually within one to two weeks in the more moderately impaired patients, to pharmacologic therapy, drawing from medications that address the three primary pathophysiologic problems in orthostatic intolerance: defective vasoconstriction/excessive dependent venous pooling, low circulating blood volume, and excessive sympathoadrenal responses to reduced cerebral blood flow when upright. The three main treatment categories are, therefore, medications to (1) improve vasoconstriction, (2) improve blood volume, and (3) control heart rate and catecholamine release or effect. We initiate treatment with one medication at a time to avoid confusion about any potential adverse effects. We stop medications that are ineffective but usually continue those that have partial efficacy, often adding medications from a different category (for example, using a vasoconstrictor to improve venous return to the heart in combination with a beta-blocker to reduce heart rate). 

Case Report

The patient increased his oral sodium chloride intake and used lower limb compression garments. Table 2, Table 3 and Table 4 show the medications and supplements prescribed for this individual, and Figure 2, Figure 3 and Figure 4 illustrate the timing of their introduction. He was unresponsive or had adverse effects with fludrocortisone, midodrine, atenolol, pyridostigmine bromide, and ivabradine. He reported improvements associated with low doses of clonidine and methylphenidate, noting that if he missed doses of these medications, he felt worse in the next 24 h. 

### 3.2. Comment: Allergic and Mast Cell Inflammation

Some forms of POTS have been associated with mast cell activation syndrome [32] (MCAS), and POTS has been identified as part of the phenotype of hereditary forms of tryptasemia [33]. Syncope and other forms of orthostatic intolerance are associated with mastocytosis and with aberrant mast cell degranulation [34]. Furthermore, it has been hypothesized that mast cells have a pathophysiologic role in both the acute response to SARS-CoV-2 infection and in post-COVID conditions [35,36,37]. Some preliminary work suggests that medications used for treating MCAS, such as H1 and H2 antihistamines, may have benefits for long COVID [38]. It is unclear how many patients with ME/CFS also have MCAS, but there is a substantial overlap in presenting symptoms [13]. 

Among the more recognizable features of MCAS are recurring rashes, facial flushing, urticaria, and intolerance of several foods and medications. There is vigorous debate about the most appropriate criteria and laboratory tests to confirm the diagnosis, reviewed in Diagnosis of mast cell activation syndrome: a global “consensus-2” [30]. Serum tryptase is usually normal in MCAS except among patients with hereditary alpha-tryptasemia [33]. Many laboratory tests for MCAS are temperature-sensitive and are, therefore, prone to handling errors. Some diagnostic criteria include the response to treatment as supporting the diagnosis. Recommended treatments include avoiding specific dietary and environmental triggers to mast cell activation, then adding medications and supplements capable of blocking histamine and other mast cell mediators or stabilizing the mast cell membrane. 

In our patient, there was a long history of allergic inflammation. He had been treated with immunotherapy from ages 10–13 years for allergic rhinitis and still experienced oral allergy syndrome symptoms (mucous membrane itching and swelling) after exposure to cherries, cashews, and carrots. He had mild asthma. Both parents had a history of allergies and prior immunotherapy. Laboratory testing on several occasions during the first year of post-COVID illness confirmed an elevated plasma histamine (3.5 ng/mL, 3.2 ng/mL, 2.0 ng/mL, with a normal range of ≤1.8), as well as a mild elevation in chromogranin A (144 ng/mL, with a normal range of 25–140), which can be released by mast cells [30]. We interpreted these results as consistent with mast cell activation syndrome, using the “consensus-2 definition” [30,39]. This definition requires the presence of symptoms of increased mast cell activity along with one of the following minor criteria: (1) multifocal or disseminated infiltrates of mast cells in marrow and/or extracutaneous organ(s), (2) abnormal spindle-shaped morphology in >25% of mast cells in marrow or other extracutaneous organs, (3) abnormal mast cell expression of CD2 and/or CD25, (4) genetic changes shown to increase mast cell activity, (5) above-normal levels of tryptase, histamine or its metabolites, heparin, chromogranin A, or other mast cell mediators, (6) symptomatic response to inhibitors of mast cell activation or mast cell mediator production or action. Our patient met the major criterion as well as minor criteria 5 and 6. We did not test for the other criteria as these were deemed unnecessary due to the cost and time involved, especially since he already clearly met the diagnosis. 

We treated him with the medications shown in Table 3; Figure 3 illustrates the time course of their introduction. Of interest, stopping fexofenadine and famotidine was accompanied on more than one occasion by increased fatigue and increased cognitive dysfunction, both of which improved rapidly upon resumption of treatment. The addition of cromolyn was associated with an abrupt improvement in peak heart rate when walking; his heart rate fell from the 130–140 bpm range to 100–105 bpm on cromolyn. 

We added escitalopram to address the increase in sensory overstimulation that had appeared after COVID, with a background of personal and family history of mild anxiety. We added naltrexone, which has some utility for addressing symptoms in both fibromyalgia and ME/CFS [40,41,42]. This was associated with an improvement in energy and overall function (Table 4 and Figure 4). 

### 3.3. Comment: Naltrexone

In a retrospective study of 218 patients with ME/CFS, naltrexone was associated with self-reported improvement of symptoms in 73.9%, most commonly in alertness, physical performance, and cognitive dysfunction [40]. Cabanas and colleagues found that the impaired function of Transient Receptor Potential Melastatin 3 (TRPM3)—nociceptor channels thought to be involved in the pathophysiology of ME/CFS—was restored by naltrexone [41]. 

Even with these medications and supplements, for the first 15 months after the acute infection, he continued to have exercise intolerance, characterized by a heart rate that would reach 130–150 bpm during each of two 15-min walks daily. This peak heart rate suggested to us that he was not yet ready for further increases in his activity level and that more activity might provoke PEM. While both our clinical impression and the report from the patient suggested that each of the interventions brought at least a modest benefit, we clearly cannot exclude the possibility that some of the apparent improvement could have been spontaneous as he got further away from the initial COVID infection. Furthermore, we cannot be sure of the independent effect of each intervention.

## 4. Follow-Up

During Fall 2020–Spring 2021, namely the first year of the illness, he was able to take online university courses. By the summer of 2021, he was able to tolerate a virtual summer internship that required 40 h per week of work. By July 2021, a year after his initial infection, his remaining symptoms included daily fatigue, unrefreshing sleep, and post-exertional malaise. Lightheadedness and headaches by this point were infrequent. 

By the Fall of 2021, 15 months after the acute infection, he had the stamina to attend in-person courses and to walk around his university campus. At this point, he began physical therapy to address his movement restrictions. After these had been treated, he was advanced to stationary biking, with the goal of flexible increases in the duration and intensity of activity, designed to avoid provoking post-exertional malaise. He advanced his physical activity regimen as follows:September 2021: biking 15 min twice a weekOctober 2021: biking 20 min three times a week (HR 120–140 bpm)November 2021: biking 25 min four times a weekJanuary 2022: biking 30 min four times a week

He had a sense that the 25 min four times a week was more challenging, associated with a slightly higher heart rate than with the 20-min exercise duration, so he remained at that volume of exercise for a longer period of time. After successfully managing 30 min of biking four times a week, by March 2022, he was able to resume running, starting with 10 min four times a week. He increased his running time by two minutes each week as long as he had not experienced excessive heart rate increases or post-exertional malaise. In October 2022, he was able to compete with the cross country team in an 8 K event, but with race times that were considerably slower than his pre-illness efforts. Thus, while improved, he was still not back to his pre-illness level of fitness and function.

## 5. Discussion

This case is a practical demonstration of applying our multi-disciplinary management approach to ME/CFS in a patient with long COVID. This approach involves a focus on known, common ME/CFS comorbidities. In our patient, the main comorbid problems were the range of motion abnormalities, mast cell activation, allergic inflammation, and orthostatic intolerance. Other ME/CFS comorbidities, which were not present in our patient, can include neurogenic thoracic outlet syndrome [43,44], venous insufficiency (pelvic congestion syndrome with ovarian and internal iliac vein varices, May Thurner syndrome with compression of the left common iliac vein) [45,46,47], gastrointestinal motility disturbances [48,49,50], migraine headaches [51,52], Ehlers-Danlos syndrome or non-syndromic joint hypermobility [17,53,54,55,56,57,58,59], and neuroanatomic abnormalities (Chiari malformation [60], cervical spinal stenosis [14], atlantoaxial instability [22] or craniocervical instability [23]). It is reasonable to assume that some proportion of patients with long COVID will also have had some of these comorbid conditions preceding their acute SARS-CoV-2 infection. Awareness of these issues can help guide the evaluation and treatment of both ME/CFS and long COVID. 

Afrin and colleagues have hypothesized that pre-existing MCAS might predispose patients to develop long-COVID after a SARS-CoV-2 infection by contributing to a hyperinflammatory response [35]. Our patient had not been diagnosed with MCAS prior to the acute infection but had a strong history of allergies, and his plasma histamine levels were elevated on several occasions, consistent with that diagnosis. Afrin and others have proposed that in those with unrecognized and untreated MCAS, the SARS-CoV-2 virus has the potential to activate mast cells through a variety of mechanisms, possibly including binding to the mast cell angiotensin-converting enzyme 2 receptors [35,36]. If mast cells become inappropriately hyperactivated, their release of cytokines and downstream activation of other elements of the immune system is hypothesized to promote a post-infectious inflammatory syndrome that contributes to the symptoms of long COVID [37]. In support of this hypothesis, medications used in the treatment of MCAS have been identified as potentially beneficial in treating long COVID. In one study, a combination of loratadine (or fexofenadine) and famotidine (or nizatidine) reduced the symptom burden in 72% of patients [38]. The mechanism of action is not known for all of these medications, however, some H1 antagonists exhibit ACE2 inhibitory activity, while some have direct anti-viral activity [61]. Famotidine might improve symptoms by acting as an antagonist or inverse agonist for H2 histamine G protein-coupled receptors (GPCR) [62]. In an animal model, mast cells degranulated after a SARS-CoV-2 challenge, and the degranulation was attenuated using the mast cell stabilizer cromolyn or H1 antihistamines ketotifen, ebastine, and loratadine [63]. In our patient, the H1 and H2 antihistamines were associated with improvement in cognitive function; stopping them was associated with an abrupt worsening in concentration and short-term memory, which improved upon the resumption of the medications. Mast cell stabilizers, such as quercetin and cromolyn, were associated with improvements in fatigue and temporally associated with improvements in his heart rate. Orthostatic intolerance syndromes are common in those with mast cell activation [34], and some forms of POTS have been described in association with mast cell activation disorders [32].

New post-COVID orthostatic intolerance was a significant component of our patient’s activity limitations and reduced quality of life. Major gains in his tolerance of activity were not made until the orthostatic intolerance improved. We caution that rigid advancement of exercise can provoke post-exertional symptom exacerbations in those with ME/CFS and long COVID. Our experience has been that it is critical to treat orthostatic intolerance before advancing aerobic activity and that such advances need to be flexible and conducted in a manner designed to avoid the provocation of PEM. Although several medications directed at his POTS and MCAS were temporally associated with improvements, we cannot be sure that the improvements were caused by the medications. We emphasize that the medications prescribed for this patient are not necessarily appropriate for all with long COVID. As this case illustrates, patients may need to try multiple different medications from different classes before finding an acceptable fit. There remains some uncertainty about the efficacy of specific medications directed at orthostatic intolerance, and full confirmation of the efficacy and effectiveness in the setting of long COVID awaits subsequent randomized trial evidence. However, we feel that these clinical observations have the potential to provide some guidance about treatment during a time of uncertainty and serve as an alternative to therapeutic nihilism.

## Figures and Tables

**Figure 1 healthcare-11-00865-f001:**
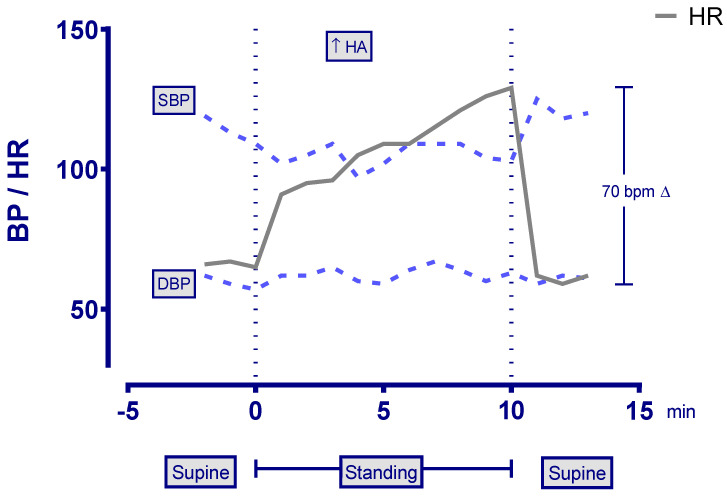
Heart rate, blood pressure, and symptom responses to a 10 min passive standing test. Abbreviations: HA—headache; HR—heart rate; SBP—systolic blood pressure; DBP—diastolic blood pressure; BPM—beats per minute. Modified with permission from Petracek, L.S., Suskauer, S.J., Vickers, R.F., Patel, N.R., Violand, R.L., Swope, R.L., Rowe, P.C. Adolescent and Young Adult ME/CFS After Confirmed or Probable COVID-19 [4].

**Figure 2 healthcare-11-00865-f002:**
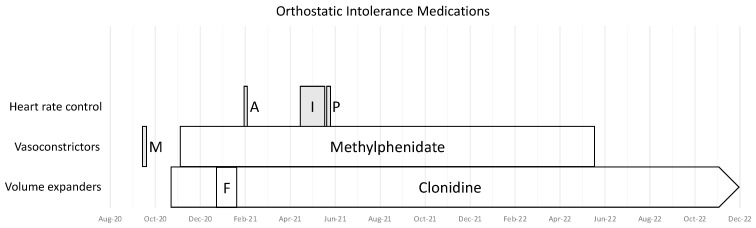
The time-course for trials of medications directed at orthostatic intolerance from August 2020–December 2022. Abbreviations: POTS—postural tachycardia syndrome; I—ivabradine; P—pyridostigmine bromide; M—midodrine; F—fludrocortisone; A—atenolol.

**Figure 3 healthcare-11-00865-f003:**
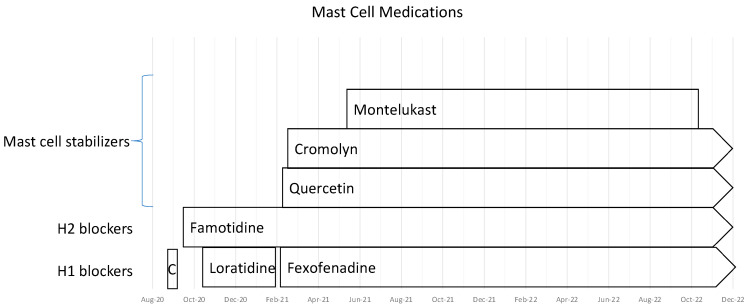
The time-course for trials of medications directed at mast cell activation syndrome from August 2020–December 2022. Abbreviations: H1—histamine 1; H2—histamine 2; C—cyproheptadine.

**Figure 4 healthcare-11-00865-f004:**
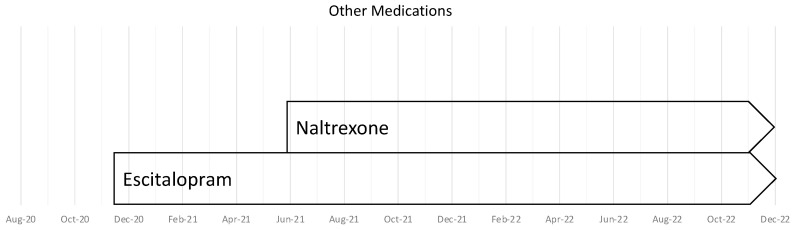
The time-course for trials of other medications from August 2020–December 2022.

**Table 1 healthcare-11-00865-t001:** Laboratory test results one month after the onset of illness * Abbreviations: WBC—white blood cells; ESR—erythrocyte sedimentation rate; TSH—thyroid stimulating hormone.

Test	Patient Results	Reference Range
WBC	5.1 K/uL	4.5–11 K/uL
Hemoglobin	16.1 g/dL	13.2–17.1 g/dL
Platelet count	255 K/uL	150–350 K/uL
Comprehensive metabolic panel	Normal except **total bilirubin 1.5 mg/dL** (consistent with a prior diagnosis of Gilbert syndrome)	Bilirubin 0.2–1.1 mg/dL
Creatine kinase	79 U/L	44–196 U/L
ESR	6 mm/h	0–22 mm/h
Plasma histamine	**3.5 ng/mL**	≤1.8 ng/mL
Serum tryptase	5.4 mcg/L	<11.0 mcg/L
Chromogranin A	**144 ng/mL**	25–140 ng/mL
TSH	1.21 mIU/L	0.50–4.30 mIU/L
T4 free	1.3 ng/dL	0.8–1.4 ng/dL
Ferritin	180 ng/mL	38–380 ng/mL
Lyme antibody screen	negative	

* abnormal values are in bold.

**Table 2 healthcare-11-00865-t002:** Medications directed at orthostatic intolerance. The ones in bold were maintained.

Problem Addressed	Medication and Dose	Response
POTS		
Volume expanders	Fludrocortisone 0.1 mg	Increased fatigue, lower mood
	**Clonidine 0.1 mg**	**Improvement in sleep and cognitive function, but higher doses were associated with increased fatigue**
Vasoconstrictors	Midodrine 7.5 mg	Ineffective
	**Methylphenidate 10 mg**	**Helpful for cognitive function, but higher doses were associated with increased insomnia. Discontinued without return of problems at 2 years after onset of illness**.
Heart rate control	Atenolol 25 mg	Improved tachycardia, but increased headaches, fatigue, and problems with concentration
	Pyridostigmine bromide	Increased fatigue
	Ivabradine 5 mg 2× day	Increased fatigue; excessive nocturnal bradycardia

Bold signifies which medications were maintained as part of the treatment regimen.

**Table 3 healthcare-11-00865-t003:** Medications directed at MCAS and their effects. Abbreviations: MCAS—mast cell activation syndrome.

Problem Addressed	Medication and Dose	Response
MCAS		
H1 blockers	Loratadine 10 mg	Helpful with energy and cognitive dysfunction
	Cyproheptadine	Ineffective
	**Fexofenadine 180 mg**	**Helpful with energy and cognitive dysfunction**
H2 blockers	**Famotidine 40 mg 2× day**	**Helpful with energy and cognitive dysfunction**
Mast cell stabilizers	**Oral cromolyn 400 mg 4× day**	**Improvement in resting heart rate**
	Montelukast 10 mg	Unclear benefit. Stopped to assess whether contributing to negative mood
	**Quercetin 1000 mg 2× day**	**Improvement in fatigue and allergic symptoms**

Bold signifies which medications were maintained as part of the trreatment regimen.

**Table 4 healthcare-11-00865-t004:** Other medications directed at symptoms and their effects.

Problem Addressed	Medication and Dose	Response
Other medications		
Sensory sensitivity	**Escitalopram 5 mg**	**Improvement in response to stressors and sensory stimuli at a dose of 5 mg daily; increased fatigue on 10 mg**
	**Naltrexone 4.5 mg**	**Improvement in energy and overall function**

## Data Availability

All relevant data are reported in the paper.

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
