# Peer review of "A Case Study of Successful Application of the Principles of ME/CFS Care to an Individual with Long COVID"

_healthcare, 2023, doi:10.3390/healthcare11060865_

Round 1

Reviewer 1 Report

The case report by Petracek and colleagues highlighted the multi-disciplinary management approach to ME/CFS in a patient with long COVID. This is a very interesting and relevant study since Long COVID remains to be one of the worst sequelae of the COVID-19 disease. The case report is also well-written, although I just want to clarify if this patient was vaccinated or even boosted previously and as to which variant this patient was infected with. There are several factors contributory to Long COVID and these should be highlighted in the introduction,

DOI: 10.1073/pnas.2024358118 DOI: 10.1007/s13365-021-01002-x DOI: 10.3390/v14122629 DOI: 10.3390/jcm11247314

Author Response

We thank the reviewer for the constructive comments. 

In the revised manuscript we have been clearer in our language about this young man's immunization status, and have clarified that he became ill in the period before the CVOID-19 vaccines were available, so he had not been  immunized at the time of infection. We also believe that he had the historical variant, not one of the newer variants, as he got sick in June 2020, before the waves of variant infections began in the US.

We have added some of the suggested references to the introduction, but as this was not intended to be a full discussion of risk factors for Long COVID, we elected not to elaborate further on that topic. 

Reviewer 2 Report

Dear Editor

This article reported the successful application of the principles of ME/CFS care to an individual with Long Covid, which is an interesting and important topic today. But there are some concerns in this article that need to be addressed.

1.     Please change Long Covid to Long COVID.

2.     Please update the references in the below sentence:

“Other ME/CFS comorbidities, which were not present in our 290 patient, can include neurogenic thoracic outlet syndrome39,40, venous insufficiency (pelvic 291 congestion syndrome with ovarian and internal iliac vein varices, May Thurner syndrome 292 with compression of the left common iliac vein)41-43 , gastrointestinal motility disturb- 293 ances44-46, migraine headaches47,48, Ehlers-Danlos syndrome or non-syndromic joint hyper- 294 mobility13,49-55, and neuroanatomic abnormalities (Chiari malformation56, cervical spinal 295 stenosis10, atlanto-axial instability18 or cranio-cervical instability19).”

3.     For better understanding, please explain TSH, WBC, and ESR below the table 1.

4.     For better understanding, please explain POTS and MCAS below the table 2.

5.     Please include the article in the manuscript:

- Hunt, Joanne, Charlotte Blease, and Keith J. Geraghty. "Long Covid at the crossroads: Comparisons and lessons from the treatment of patients with myalgic encephalomyelitis/chronic fatigue syndrome (ME/CFS)." Journal of Health Psychology 27.14 (2022): 3106-3120.

6.     Please include an appropriate reference at the end of sentence.

“Afrin and others have proposed that in those with unrecognized and untreated MCAS, the SARS-CoV-2 virus has the potential to activate mast cells through a variety of mechanisms, possibly including  binding to the mast cell angiotensin converting enzyme 2 receptors”

7.     Please include the ethical approval section (name of the organization that approved the study and ethical code).

8.     Please include consent to participate and funding (grant ID) sections.

Author Response

We thank the reviewer for this careful reading of the manuscript. 

Item 1: We have made the changes to Long COVID throughout.

Item 2: The references in this section include the original observations and the important historical observations about these co-morbid conditions, as well as some modern references. 

Item 3: Explanations for TSH, ESR, and WBC are now included as requested. 

Item 4: POTS and MCAS are now included in the Abbreviations. 

Item 5: The paper by Hunt and colleagues has been included. Thank you for this suggestion. 

Item 6: We have added references to this sentence, and added an additional new reference from Theoharides about mast cells and COVID.

Item 7: We have added a statement at the end about the ethical issues. This case report was exempt from IRB approval, and the patient gave full consent to have his information included. 

Reviewer 3 Report

A very timely and necessary piece, thank you for your vital contribution. We need more case studies such as this, and yours is a very appropriate one which importantly analyses interevention and what the 'magic ingredient' may be for some patients with LC. With your paper, I did got a clear sense that this was a treatment approach made up of a constellation of interventions. As a reader, it would have also been helpful to get a sense of whether this patient used compression and salt/fluid intake as a treatment modality also? 

It would also to have been helpful to have further post intervention HR meaurements to be entirely sure that his recovery was due to POTs . For example, he seems to take a huge leap from March 22 (running 4x a week for 10 mins) to an 8km run in Oct 22. Something has happened in this time and it would be helpful to have HR information if you have it, to contextualise why this significant recovery happened (i.e was it POTs , was it just great pacing and fatigue management, was it avoiding PESE?)

Author Response

We thank the reviewer for these comments. We had neglected to specify that the patient was using compression garments and adding sodium chloride, so have added a sentence to this effect to the revision. 

The reviewer asks whether we have further post-intervention heart rate measures. We did not perform a repeat standing test, but the patient himself tracked his heart rate as one of the ways of avoiding post-exertional malaise. We have added a mention of this to the revised manuscript. We believe the recovery was due to a mix of the factors the reviewer mentions (the therapies we provided, good pacing, avoidance of PEM, and probably some spontaneous improvement as he got further away from the initiating infection.) I think the revised manuscript is clearer about these issues.